# Multi-objective optimization of wear parameters of hybrid composites (LM6/B$_4$C/fly ash) using grey relational analysis

Charles Sarala Rubi[1], Jayavelu Udaya Prakash[2], Sunder Jebarose Juliyana[2], Sachin Salunkhe[3,4*], Robert Cep[5], Emad Abouel Nasr[6]

1 Department of Physics, Vel Tech Rangarajan Dr. Sagunthala R&D Institute of Science and Technology, Chennai, India, 2 Department of Mechanical Engineering, Vel Tech Rangarajan Dr. Sagunthala R&D Institute of Science and Technology, Chennai, India, 3 Department of Biosciences, Saveetha School of Engineering, Saveetha Institute of Medical and Technical Sciences, Chennai, India, 4 Faculty of Engineering, Department of Mechanical Engineering, Gazi University, Ankara, Turkey, 5 Department of Machining, Assembly and Engineering Metrology, Faculty of Mechanical Engineering, VSB-Technical University of Ostrava, Ostrava, Czech Republic, 6 Department of Industrial Engineering, College of Engineering, King Saud University, Riyadh, Saudi Arabia

* sachinsalunkhe@gazi.edu.tr

## Abstract

### Background

Aluminium based composites with hybrid reinforcement hold significant potential to replace Al-alloys in a variety of automotive sectors where cheap cost, a significant ratio of strength to weight, and better wear resistance are required.

### Methods

Stir casting was utilized to make aluminium matrix composites (AMCs) with 3%, 6%, and 9% of B$_4$C/Fly ash particles. The wear was examined with various Sliding Speed, S (1 m/s, 1.5 m/s and 2 m/s), Sliding Distance, D (500 m, 1000 m and 1500 m), applied load, L (15 N, 30N and 45 N) and reinforcement %, R (3, 6 and 9%). Grey Relational Analysis was used to optimise the wear variables. Taguchi's L$_{27}$ Orthogonal array (OA) was selected for this statistical approach in order to analyse responses like Specific wear rate (SWR) and Coefficient of Friction (CoF). Furthermore, analysis of variance (ANOVA) was utilized to investigate the influence of input parameters on wear behavior by choosing "smaller is better" feature.

### Results

Based on this study, the optimal values of S – 1.5 m/s, D – 500 m, L – 30 N, and R% – 9 wt% Hybrid (4.5% Fly ash and 4.5% B$_4$C) are found to yield the lowest SWR and CoF. Wear rate of composite decreased with an increase in reinforcement particles. Increase in hardness was also the reason for decrease in wear rate. The responses

**Data availability statement:** All relevant data are within the manuscript.

**Funding:** The authors present their appreciation to King Saud University for funding this research through the Ongoing Research Funding program (ORF-2025-164), King Saud University, Riyadh, Saudi Arabia. This article was co-funded by the European Union under the REFRESH – Research Excellence For Region Sustainability and High-tech Industries project number CZ.10.03.01/00/22_003/0000048 via the Operational Programme Just Transition and has been done in connection with project Students Grant Competition SP2024/087 Specific Research of Sustainable Manufacturing Technologies "financed by the Ministry of Education, Youth and Sports and Faculty of Mechanical Engineering VŠB-TUO. The article has been done in connection with the project Students Grant Competition SP2024/087", Specific Research of Sustainable Manufacturing Technologies "financed by the Ministry of Education, Youth and Sports and Faculty of Mechanical Engineering VŠB-TUO".

**Competing interests:** The authors have declared that no competing interests exist.

have a narrow margin of error, according to confirmation studies. There exists a good agreement between them.

## Discussion

The research on LM6/B$_4$C/fly ash composite fabrication using Grey Relational Analysis (GRA) has significantly contributed to the development of high-performance materials for wear-related applications. Through the optimization of wear parameters, GRA allows for the improvement of wear resistance, strength, and sustainability.

---

## Introduction

Composites are structural materials made up of multiple components that combine at a macroscopic level, have different forms and/or compositions, and are insoluble in one another. The discontinuous component is reinforcement and the one in which it is dispersed is called the matrix. The matrix is the continuous phase. These substances can be made by combining two or more distinct elements in a way that makes them work together mechanically. These materials could contain a soft phase incorporated into a hard phase, or the other way around [1]. Metal-matrix composites (MMCs) with hard dispersoids have drawn more and more interest. Using either powder metallurgy or liquid metallurgical methods, dispersoid particles such as alumina, fly ash, silica, silicon carbide, and zirconia are added to aluminum-silicon alloys. In addition, aluminium alloys are enhanced tribologically by adding soft phases such as coconut shell char, mica, and graphite [2]. Just like graphite does in cast irons, these soft particles function as solid lubricants. The primary challenges in manufacturing composites with soft particles encompass porousness, poor durability, and the particles' wettability due to the matrix [3]. Wear is an utmost phenomenon carried on by material dislocation and separation. Variations in size and a gradual decrease of weight over time are implied by wear. Every mechanical part that comes into contact with something sliding or rolling will eventually wear out. These parts include splines, brakes, clutches, piston rings, seals, guides, gears, and bearings. The size, velocity, and composition of the dispersoids have a significant influence on how AMCs wear. If the volume of the ceramic particle is more and the dispersoid is firmly bound to the continuous phase, the composite wear rate steadily drops. The load used in the wear test determines this essential volume percentage [4]. When compared to pure aluminium alloys, AMCs generally have greater mechanical strength and hardness. Furthermore, it has been shown that using AMCs can greatly improve tribological characteristics, especially in terms of preventing slippage, abrasive wear, and seizure [5]. The diffusion of material from one layer to another while movement between them as a consequence of local adhesion between the contacting surfaces or as a result of solid-phase welding is known as adhesive wear [6]. When particles are extracted from one surface, they can bond to the other surface indefinitely or momentarily. When surfaces glide against one another and there is enough pressure across the contacts imperfections to produce limited plastic distortion, adhesive wear results.

The actual area of contact that exists among imperfections of contacting materials is established by a material's hardness. For this reason, asperity hardness is valued more than bulk hardness [7]. A type of mechanical wear known as abrasive wear develops when a tougher material repeatedly slides or hits a metal surface, wearing away the material in the process. In manufacturing environments, where metallic elements are subjected to coarse particles like sand, grit, or mineral ores, this wear phenomena are commonly observed. Similar to tiny cutting instruments, these abrasive particles cause microfractures, cracks, as well and distortion on the metal surface, which eventually results in material loss and a shorter component lifespan [8]. As it depends on various elements, such as operational circumstances and the disc's attributes, the resistance to abrasion should not be uniformly regarded as an intrinsic attribute of the material [9]. The qualities of the abrasive material and its counterpart, the size and composition of the dispersoid, the 'S' & 'L' and other factors all affect how much abrasive wear occurs. Higher strength, toughness, and hardness levels in materials usually reflect into better resistance to wear. However, even these substances are subject to significant deterioration that require expensive replacements or maintenance. Several techniques, such as coating, hardening, or using wear-resistant alloys, have the potential to reduce abrasive wear. Higher toughness, resilience, and rigidity levels in materials usually translate into better endurance against abrasive wear. However, even these substances may get severely worn out and require expensive repairs or substitutes [10]. In comparison to Al-alloy, AMCs have better wear and deformation resistance. This is because the reinforcement increases the virgin alloy's high-temperature strength and cause the continuous phase to mechanically restrict. The hard dispersoid protects the matrix from the counter-face, which lowers the wear and CoF in the composite relative to the alloy [11]. By avoiding direct metal contacts, the subsurface deformation is avoided, increasing the composites' resilience to wear. It is demonstrated that the use of hard ceramics raises resistant to wear. The matrix alloy's mechanical properties improve when a dispersoid is added. The inclusion of ceramic particles results in increased abrasion resistance, sliding wear resistance, and a delayed transition from moderate to substantial wear [12]. Reinforcement particles with superior bonding with the continuous phase withstand the load and restrict the initiation of cracks. This phenomenon helps to improve the wear resistance (WR) [13]. The MMCs' WR was considerably greater compared to that of the Al alloy, and it raised as the amount and size of alumina particles produced while decreasing when the 'D', and 'L' increased. Particle density has less impact on wear resistance than the size of the Alumina particles [14]. The composites' resistance to abrasive wear increased as the size and concentration of $B_4C$ particles increased [15]. In MMCs, the WR depends on the type and amount of dispersoid used. Initial intense WR is more effectively prevented by fibers and particles than whiskers [16]. Particulates are more advantageous for enhancing the WR of the MMCs. Ceramics such as SiC, $Al_2O_3$, $B_4C$, TiC and $TiB_2$ are commonly used dispersoid of aluminium [17]. Because of its excellent chemical durability, outstanding strength, light weight, $B_4C$ is considered to be one of the most desirable materials for ceramics. $B_4C$ can be utilised as a complement to SiC and $Al_2O_3$ for strengthening AMCs, especially in situations where good wear resistance is crucial. Due to the $B_{10}$ isotope's capacity to absorb neutrons, aluminum/$B_4C$ composites are used in the nuclear industry [18]. Ramachandra & Radhakrishna et al [19] fabricated AMCs with Al as continuous phase and Si & Fly ash as dispersoids (with 12 and 15 wt% respectively) using the stir casting route. They used various wear tests to examine the wear properties of composites in as-cast circumstances. The optical microscope and SEM have been used to study the worn surfaces. The findings show that while the WR of the composite reduced with raising 'L' and sliding velocity, it raised with an increase in dispersoid. Rohatgi et al [20] analyzed the abrasive wear features of A356 Al alloy with fly ash composites. The findings show that, for loads up to 8 N, the composite material has good abrasive resistance to wear than the matrix alloy and is comparable to the alumina fibre reinforced composite. The detaching and shattering of dispersoids lower the WR of the composite for loads greater than 8 N. Using a stir-casting process, Ravi Kumar et al. [21] created Al alloy composite materials strengthened with fly ash particulates of 3 distinct sizes in 3, 6, 9, and 12 weight %. For a fixed duration of ten minutes, wear tests were done on pins and discs with three distinct weights and 'S'. They constructed an equation for estimating the composites' CoF and wear. The created model's validity was additionally examined with ANOVA approach. When it comes to WR, MMCs reinforced with coarse fly ash particles outperform those with tiny

particles. Researchers have reported that the WR of the composites increased on raising the % of dispersoid [22]. The WR of the MMCs can be increased by as much as 70% by raising the percentage of ceramics. It was additionally determined that when the amount of particles volume fraction rises, so does the dry sliding resistance to wear. The adhesive wear resistance depends on the amount of $B_4C$ present. Raising $B_4C$ above 15% does not significantly increase wear resistance under the test conditions because of the debris collection [23]. It emerged that the wear volume reduced with raise in the filler material volume and increased with a raise in the 'L'. This could be because there was an apparent reduction in ductility after the inclusion of ceramic particles [24]. Furthermore, a rise in hardness could potentially enhance the composites' resistance to wear. It was observed that when SiC was added in greater amounts, the wear rate dropped at any constant load and the Al-alloy's load-bearing capabilities were enhanced while sliding. SiC was added in increasing levels, which limited the matrix material's ability to flow or deform in response to load [25]. WR rises with an increase in dispersoid content and hardness. The main issue regarding AMCs is that, both the fine size and the higher volume percentage of the dispersoid particles make them even more expensive. As a result, reducing or eliminating the usage of smaller particles and controlling the proportion of volume are vital steps towards lowering the expense component. As the volume proportion of dispersoid increased, the composites' abrasive wear properties reduced more quickly. The mean CoF declines with a rise in 'L' and percentage of SiC, while WR drops linearly with a rise in SiC weight fraction [26]. The interfacial adhesion among the continuous phase and the dispersoid is an indication of the wear characteristics ceramic dispersed composite. This is because less material wear is produced by the robust interfacial bond, which is essential in distributing load from the continuous phase to the dispersoid [27]. The research on LM6/$B_4C$/fly ash composite fabrication using Grey Relational Analysis (GRA) has significantly contributed to the development of high-performance materials for wear-related applications. Through the optimization of wear parameters and process conditions, GRA allows for the improvement of key properties such as wear resistance, strength, and sustainability. The use of GRA ensures that the composites are designed for optimal performance in automotive, aerospace, engineering, and industrial applications, making them an ideal choice for demanding wear environments. Although Grey Relational Analysis (GRA) has been used in wear studies, its applicability to hybrid composites such as LM6/$B_4C$/Fly ash has not been thoroughly investigated in prior research. Our approach stands out due to its thorough experimental validation and simultaneous optimization of several wear factors. To provide a more reliable framework for wear optimization in hybrid composites, we also improve the GRA technique to increase its accuracy in forecasting wear behavior.

## Materials and methods

Based on factors like cost, use, and qualities, the materials used in this investigation were selected. Three sets of plates measuring 100 mm by 100 mm by 10 mm with reinforcement weight percentages of 3, 6, and 9, accordingly, were developed by stir casting technique [28]. Boron carbide is one of the hardest materials, which significantly improves the wear resistance of the LM6 matrix, making the composite suitable for applications involving abrasive environments. It can enhance the tensile and compressive strength of the aluminium alloy, contributing to higher stiffness and durability of the composite [29]. Incorporating fly ash into the LM6 alloy helps in waste utilization and reduces the environmental impact of landfill disposal of fly ash. The presence of fly ash, enhance the mechanical properties of the LM6 such as impact resistance and compressive strength. Fly ash is light in weight, helping to maintain the low density of the aluminium matrix. So, the incorporation of $B_4C$ and fly ash into the LM6 matrix results in hard, wear resistant, light weight, cost effective and environmentally friendly composite material, making it suitable for various advanced engineering applications in automotive, aerospace and construction industries.

### Materials

**LM6 alloy.** Aluminium alloys, such as LM6 alloy, are challenging to process because of their propensity to drag and their high Si content, which accelerates tool wear. The LM6 alloy has remarkable resistance to corrosion in both typical marine and atmospheric environments [30]. Table 1 displays the composition of the LM6 alloy.

**Table 1. Composition of LM6 alloy.**

| Constituent | Si | Cu | Fe | Mg | Mn | Ti | Ni | Zn | Al |
|---|---|---|---|---|---|---|---|---|---|
| Weight % | 11.48 | 0.013 | 0.52 | 0.02 | 0.01 | 0.02 | 0.01 | 0.01 | Remainder |

**Boron carbide ($B_4C$).** For this investigation, 63-micron-sized $B_4C$ particles were employed as one type of second phase material. $B_4C$ is widely employed as cermets and armour materials due to its many desirable qualities [31]. In a nuclear reactor, components containing materials with a high neutron absorption cross-section are used as control rods, so, boron is very well suited. Fig 1 depicts the Surface structure of the $B_4C$.

**Fly ash.** It is another second phase material with particle size of 12-microns. The resistance to wear, damping qualities, hardness, stiffness, and density of LM6 alloys are all improved by the inclusion of fly ash. Given their low cost and low density, fly ash particles, a waste by-product have the capability to be discrete fragments utilized in metal matrix composites [32]. They are also readily available in large quantities. Table 2 displays the fly ash composition. Fig 2 depicts the fly ash particle shape.

## Preparation of hybrid composite materials

The LM6 alloy ingots are gradually heated to 850 degrees Celsius. The melt has been degassed at 800° C using hexa-chloroethane. Fly ash and $B_4C$ particles that have been heated to 250 degrees Celsius were added to the molten metal after it has been agitated to form a vortex. For ten minutes, the slurry was agitated at 600 rpm. $K_2TiF_6$, 1% by wt was added to increase the wettability and interfacial bonding of the LM6/$B_4C$ composites in order to generate a reaction layer on an interface. Mg was added to enhance the ability to wet of fly ash [33,34]. Boron carbide and fly ash were added at wt. % of 1.5, 3 and 4.5% respectively. After being agitated and distributed, the slurry was poured in to 650 degrees Celsius preheated mould and allowed to cool. Fig 3 displays the stir casting equipment utilized during the manufacturing technique. The experimental study concluded that 600 rpm and 10 min is the best combination of stirring speed and stirring time for uniform distribution of $B_4C$ and fly ash particles over the Al matrix. Mechanical stirrer forms vortex and reinforcements particles are feed in the centre of the vortex. High feed rate results in particle accumulation in the composite and

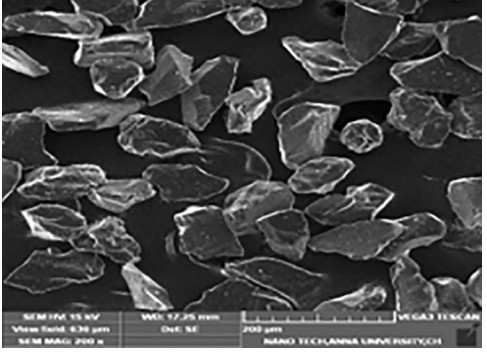

**Fig 1. Surface structure of $B_4C$.**

**Table 2. Elemental composition of Fly Ash.**

| Constituent | Al | Si | O | Fe | Ti | K | Ca | LOI |
|---|---|---|---|---|---|---|---|---|
| Weight % | 16.73 | 26.43 | 38.88 | 3.82 | 1.42 | 0.99 | 0.5 | Remainder |

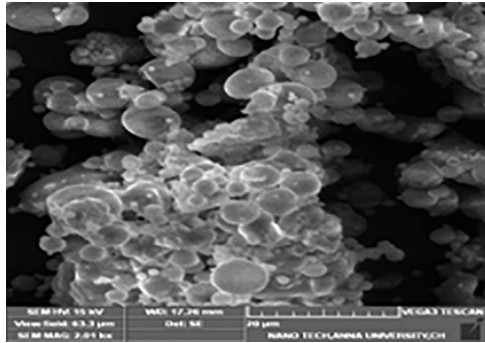

**Fig 2. Surface structure of fly ash particles.**

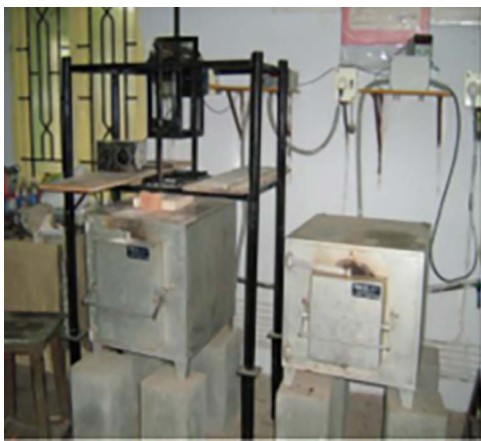

**Fig 3. Stir casting setup with furnace.**

low feed rate is difficult to achieve due to the formation of lumps of small solid particles [35]. Thus, selection of optimal rate of feeding is crucial. The optimal rate of feeding is in the range of 0.8–1.5 g/s to avoid the accumulation of reinforcements in the composite and achieve homogeneous dispersion of reinforcement particles throughout the composite [36].

## Design of experiments (DoE)

DoE strategy can be employed to determine what information, in what amount, and under what conditions need to be gathered during an experiment in order to meet two primary objectives: lower expenses and greater statistical reliability in the outcome variables [37]. In the current analysis, four different process parameters were selected; viz., 'S', 'D', 'L' and 'R'. The responses are SWR and CoF. The $L_{27}$ array is chosen for this research based on the chosen parameters. The remarkable feature of the $L_{27}$ array is that the 2-way interactions among the different components with distinct columns, which reduces their impact on the computation of the major influence of the different variables. Pilot experiments were carried out after detailed literature review. Based on the results obtained from the pilot experiments, the wear parameters were chosen. Because of load, sliding speed, and sliding distance have an immediate influence on tribological performance and are applicable to real-world scenarios, they were chosen as wear test conditions. The selected load range (15–45 N) reflects the forces found in industrial machinery and automobile brake systems. In automotive and aeronautical

components, the sliding speed range (1–2 m/s) is equivalent to rotational and linear motion conditions. In order to evaluate long-term wear behaviour under realistic usage conditions, the sliding distance range (500–1500 m) is used [38]. These variables were chosen because they offer valuable information about the wear properties of LM6/B$_4$C/Fly ash hybrid composites in practical engineering settings. There are four criteria and three levels to them. As a result, full factorial design has to be used in 3$^4$ = 81 experiments; however, DoE was only used 27 experiments. 1/3 experiments were carried out, saving 66% of the time and materials. For every experimental condition, three repetitions of the experiment have been conducted. Table 3 shows the machining variables together with their respective levels.

**Wear experiment of hybrid composites**

**Wear test – pin on disc.** Fig 4 displays the pin-on-disc system used to study the wear characteristics of materials made of composites and Fig 5 shows the photograph of pin and disc specimen.

**Test specimen and sample preparation.** The stir-casted composites are machined into pins with a 'd' and 'l' of 6 mm and 40 mm respectively. The surface is finished to a level of 0.5 μm. The counter face material for calculating the composites is made of AISI 4140 steel discs with dimensions of 55 mm in 'd', 10 mm in 't', and a hardness of 55 HRC [39]. These discs have been processed and reduced to a SR of around 0.5 μm.

**Test procedure.** According to ASTM G 99−05, the dry sliding wear tests were performed in a dry sliding circumstances at ambient temperature. Before testing and before weighing, the pin and disc materials were thoroughly cleaned to eliminate the impurities from the sample. Both before and after the wear test, the specimen's weight was measured with a digital weighing equipment to an accuracy of 0.0001 g. After the disc is securely installed on the holding apparatus, the pin is firmly inserted into its holder and adjusted so that the sample is aligned with the disc surface [40]. Mass is added to the assembly's lever in order to provide the chosen force that presses the pin against the disc. First, the motor was turned on and its speed was adjusted to the desired level while maintaining the pin sample away from the disc. The timer was set to

**Table 3. Input Parameters and their levels.**

| Level | Sliding Speed, S (m/s) | Sliding Distance, D (m) | Load, L (N) | Reinforcement, R (wt %) |
|---|---|---|---|---|
| 1 | 1 | 500 | 15 | 3 |
| 2 | 1.5 | 1000 | 30 | 6 |
| 3 | 2 | 1500 | 45 | 9 |

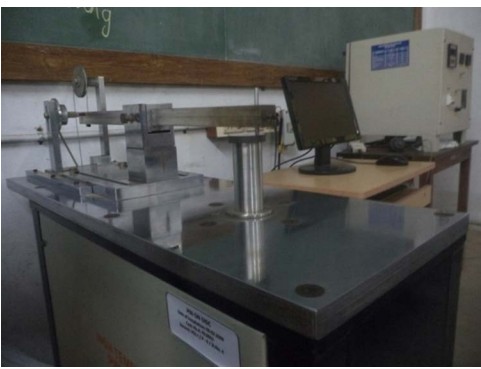

**Fig 4. Image of Pin on disc setup.**

the desired value. The specimens being tested are in contact while the test is being run. The test concludes automatically after the allotted time has elapsed.

The Volume loss, SWR and CoF are computed using the equations (1–3) respectively.

$$V = \frac{m_1 - m_2}{\rho} x\ 1000\ \ mm^3$$

(1)

$$SWR = \frac{V}{L\ x\ D}\ mm^3/Nm$$

(2)

$$CoF,\ \ \mu = \frac{F_T}{F_N}$$

(3)

Where $m_1$ is the mass of the specimen before the wear test, $m_2$ is the mass of the specimen after the wear test, $\rho$ is the density of the specimen in g/cm³, V is the volume loss in mm³, L is the applied load in N, and D is the sliding distance in m. The coefficient of friction is calculated by the ratio between tangential force ($F_T$) and the normal force ($F_N$). The tangential force is obtained from the load cell fitted in the pin-on-disk apparatus. The normal force is the applied load.

### Grey relational analysis (GRA)

Using the Taguchi approach in conjunction with GRA, the process variables have been optimised multi-objectively. The Grey Relational Theory offers an effective way to handle discrete data, multiple inputs, and uncertainty. GRA can be used to estimate the probable association among sequences because it measures the absolute value of the data variance among sequences [41,42]. In order to optimize process variables, the subsequent procedures are performed out:

(1)  Normalise the research findings for the wear test (SWR & COF).

(2)  Determine the sequence of deviations.

(3)  Determine the GRC.

(4)  Find the GRG by taking the average of the GRC.

(5)  Examine the experimental findings using statistical ANOVA and GRG.

(6)  Choose the optimal levels of process variables.

(7)  Use the confirmation experiment to confirm the optimal variables.

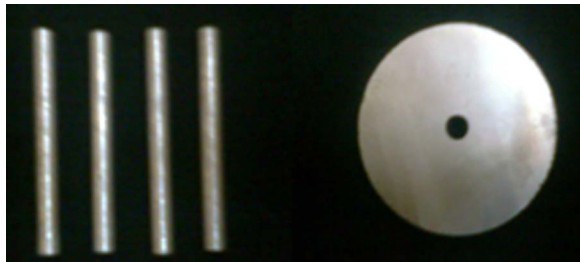

**Fig 5. Image of the prepared sample (Pins and Disc).**

The novelty of the research lies in the innovative combination of LM6, Boron Carbide (B$_4$C), and Fly Ash to fabricate wear-resistant composites, enhanced by the application of Grey Relational Analysis (GRA) for optimizing wear parameters. This research provides a unique, data-driven approach to optimizing the material composition and fabrication parameters for wear applications, combining sustainability with enhanced performance and industrial applicability. The use of GRA for multi-objective optimization is particularly groundbreaking, enabling researchers and manufacturers to predict and control wear behavior more effectively and efficiently.

## Results and discussion

The experimental results on Wear behavior of LM6/ B$_4$C/ Fly Ash AMCs are presented here. Multi-objective optimization was done using GRA with the aim of getting minimum SWR and CoF at the same time.

### Micrograph of LM6 alloy and hybrid composites

For the microstructure analysis, an optical microscope was employed. LM6 alloy and LM6 Hybrid composites were provided with mirror-like polish on their surface for testing [43]. The goal of the microscopic analysis is to confirm the homogeneous distribution of the dispersoids. In Fig 6, microscopy images are presented.

 The microscopic structures of the materials are investigated using the optical microscopic technique. The micrograph has a scale length of 50 µm, and the picture has been enlarged 200 times. In LM6 alloys, the dispersion of particles is uniform (Fig 6a). The higher silicon content in LM6 alloys maintains this distinctive uniform dispersion pattern. The lengthy, acicular, script-shaped eutectic elements of LM6 remain unaltered. The distribution of 3% in the aluminium matrix hybrid composites (1.5% fly ash and 1.5% B$_4$C) is depicted in Fig 6b. The microscopic image of a 6% hybrid composite of LM6 alloys is shown in Fig 6c. The microscope images of the 9% hybrid composites are depicted in Fig 6d. These two materials are distinguished in the matrix by their respective darkness and light colours and size of the particles are 63 & 12 microns respectively.

 During the casting process, the main objective is to avoid the formation of intermetallic phases in the composites specimens that can further degrade their interfacial properties. It has been noticed that B$_4$C has tendency to react with the Al at a higher processing temperature to form intermediate compound such as Aluminium carbide. The presence of aluminium

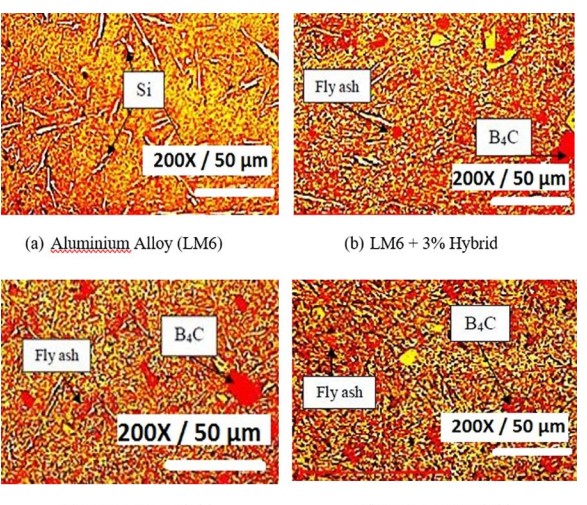

(a) Aluminium Alloy (LM6) (b) LM6 + 3% Hybrid

(c) LM6 + 6% Hybrid (d) LM6 + 9% Hybrid

**Fig 6. Optical microscopic image of B$_4$C/Fly Ash reinforced hybrid AMCs.**

carbide reduces the interfacial strength of the developed composites [44]. In order to avoid this, addition of $SiO_2$ into the Al-alloy has been found to be quite beneficial. David Raja Selvam et al [45] have found that incorporation of Fly Ash particles, which is a major source of $SiO_2$, into LM6-alloy prevents the formation of Alumnium carbide in the composite specimens.

In this study, hybrid reinforcements ($B_4C$ and FA) were reinforced into the Al-alloy using stir casting route and micro-structural analysis of developed composites revealed uniform dispersion of the particles within the composites. A clear interface between the reinforcement phase and the Al-alloy indicates that the formation of intermediate compounds was suppressed by addition of FA particles. It has been found that the Al-alloy reacts with the Mg present in the melt to form other compounds Magnesium aluminium spinel. Due to this, interfacial strength of the LM6/$B_4C$/FA composites was improved with addition of the FA particles. It has also been noticed that addition of hybrid reinforcements ($B_4C$ and FA) improved the hardness of the unreinforced LM6 alloy. This shows that the composites possess higher load bearing capacity due to presence of reinforcements [46].

## Measurement of hardness

The hardness was measured using a conventional Rockwell hardness tester from R.K. Instruments, located in Kohlapur, India. The hardness of LM6 alloy and AMCs was evaluated in the E scale. To precisely quantify the object's hardness, the samples were indented, and measurements were taken randomly at five different locations for every substance [47].

The composites' hardness is greater than the parent alloys, as Fig 7 demonstrates. It has also been found that the hardness of the composite raises with raise in the % of reinforcement [48]. By incorporating hard particles, the composites become harder and more resistant to plastic deformation [49]. Hardness increases when fly ash particles are added to cenospheres and precipitation systems [50]. Plastic deformation is prevented by such hard-reinforcing grains. R% inhibits atom dislocation, which raises the hardness, and fortifies the continuous phase. A comparable situation is reported by Mattli et al [47]. The hardness of composites increased with an increase in reinforcement content, which gives a near linear relationship with hardness. The maximum observed increase in hardness of composites compared to aluminium alloy was 37%. The observed increase in hardness was due to hard reinforcements, which act as a barrier for the movement of dislocations within aluminium and exhibit greater resistance to indentation. Furthermore, it is evident that an increase in particle size also increased the hardness. Increase in hardness may be due to the increased resistance to dislocation

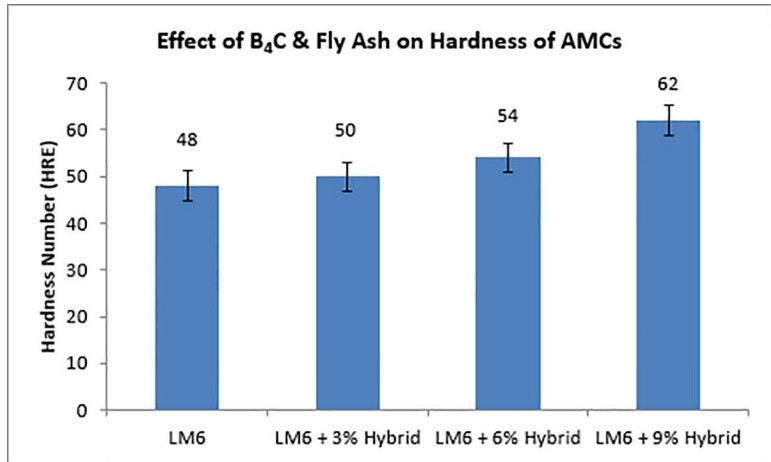

**Fig 7. Effect of $B_4C$ & Fly ash on the hardness of AMCs.**

movement by the coarse particles present in the composite. Hamouda et al. [48] and Singla et al. [49] reported that increase in addition of silicon dioxide and silicon carbide increased the hardness of aluminium alloy (LM6) and aluminium scrap, respectively. Mahendra and Radhakrishna [50] observed an increase in hardness while reinforcing fly ash particles with Al-4.5% Cu alloy produced by conventional casting technique. The presented work is in correlation with the above reported research work.

## Measurement of density

Archimedes' principle is used to determine the density of AMCs directly. The sample's mass is determined with accuracy within 0.001 g. The total volume is calculated by measuring the shifting of water using a graduated cylinder. Equation 4 is used to figure out the density of the material.

$$Density = \frac{Mass}{Volume}$$

(4)

Fig 8 illustrates how the density of AMCs decreases as the reinforcing percentage rises [51]. The density of dispersoids is lesser than that of an aluminium alloy (2.65 g/cm3), which explains the cause. According to the law of mixtures, the theoretical densities for 3%, 6%, and 9% of hybrids are 2.64, 2.63, and 2.62 g/cm3, respectively. The pattern is also shown in the quantitative density, however because of the clustering of the ceramic particulates, porosity increase as reinforcing material is added. Porosity, in terms of the material's total factors, is the quantity of pores in the material. The calculation makes use of equation 5. AMC porosity is displayed in Fig 9.

Previous studies have demonstrated similar changes in density, and their outcomes [52,53]. Monolithic $B_4C$ ceramic is a low-density material that is incredibly hard, strong, and stiff. LM6-$B_4$C-Fly ash composites can combine the aluminium ductility and the exceptionally high stiffness and outstanding hardness of $B_4$C to generate a robust low-density material [54,55]. It is clear from the density statistics that creating lightweight composites demands a decrease in density. The density of the composite material was less than that of the aluminium alloy for all reinforcement percentages. Since the density of composite is less than that of the aluminium alloy, it can be used for lightweight applications.

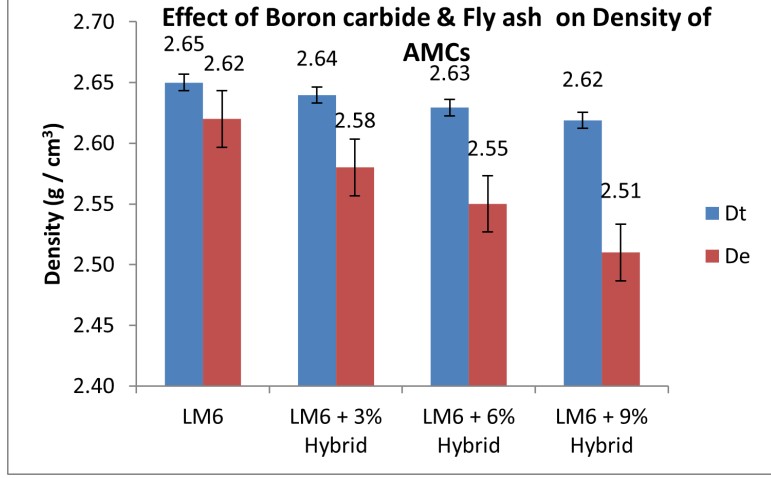

**Fig 8. Impact of $B_4$C & Fly ash on the Density of AMCs.**

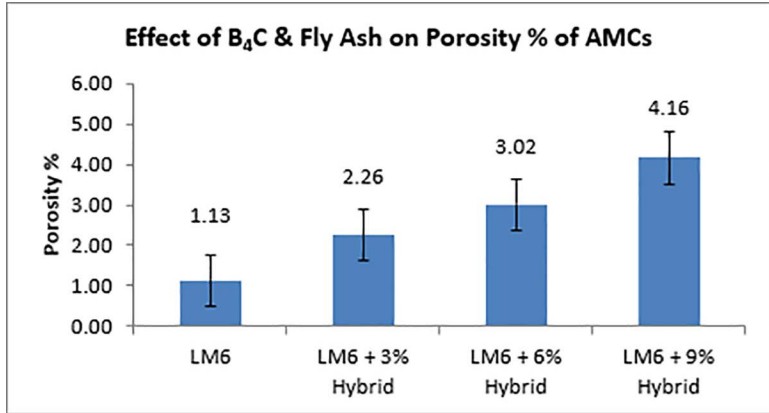

**Fig 9. Impact of dispersoids on the Porosity of AMCs.**

## Experimental results

The investigation's outcomes, their Ranks, and their GRG values are reported in Table 4.

## Analysis of the results

The Taguchi method's descriptive approach was utilized to conduct the wear test. The quantitative impacts on the outcome features are examined using the response graphs. To identify which variables are important and how much of an impact they have on the response characteristics, the GRG data are subjected to ANOVA. By examining the ANOVA tables, the best probable process parameter values are determined with regard to mean response features. Fig 10 illustrates how the GRG rises when load and reinforcement increase while sliding speed and distance decrease.

This is because of the fact that when 'S' and 'L' rise, the SWR and CoF also rise, resulting in a quicker rate of wear. As the weight % of dispersoid increases, the wear rate minimises. In addition to serving as a protective covering, a transfer layer on composites lowers friction and wear rates.

### 3.3 Selection of optimal levels

The mean of each output feature for every factor is displayed in Table 5. 'S', 'D', 'L', and 'R' are the factors that have the most impact on attaining lowest SWR and lowest CoF simultaneously. Fig 5 illustrates that the optimal values are gained at the 2nd sliding speed level, first sliding distance level, second load level, and third reinforcement level. The significance of the process components in connection to GRG is investigated using ANOVA. The F-values from the table are $F_{0.05, 2, 8} = 4.46$ and $F_{0.05, 4, 8} = 3.84$ at a five percent confidence level. Therefore, it is seen that from ANOVA Table 6, that 'D' and load, as well as the substantial relationships between 'S' and 'D' and 'D' and 'L', are important. The error term is the result of pooling the factor 'R'.

### Confirmation experiments

Both the confirmation experiment and the GRG prediction are done using the ideal parameters. The most appropriate parameters are determined by analysing the experimental data. The components at level $S_2$, $D_1$, $L_2$, $R_3$, which are the 'S' of 1.5 m/s, the 'D' of 500 m, the 'L' of 30 N, and the 'R' of 9%, can be found in Fig 10 and Response Table 5 as the ideal values for attaining low wear rate and lowest CoF. The experimental GRG is 0.907 while the expected GRG is 0.900.

**Table 4. Experimental Results of SWR and COF (LM6/ B$_4$C/ Fly Ash).**

| Ex. No | Sliding Speed S (m/s) | Sliding Distance D (m) | Load, L (N) | Reinforcement R (wt. %) | Wear Rate x 10$^{-5}$ (mm³/Nm) | COF | GRG | Rank |
|---|---|---|---|---|---|---|---|---|
| 1 | 1.0 | 500 | 15 | 3 | 9.78 | 0.408 | 0.553 | 17 |
| 2 | 1.0 | 500 | 30 | 6 | 5.12 | 0.404 | 0.668 | 8 |
| 3 | 1.0 | 500 | 45 | 9 | 10.15 | 0.334 | 0.600 | 14 |
| 4 | 1.0 | 1000 | 15 | 6 | 10.25 | 0.525 | 0.491 | 23 |
| 5 | 1.0 | 1000 | 30 | 9 | 5.08 | 0.211 | 0.898 | 2 |
| 6 | 1.0 | 1000 | 45 | 3 | 3.26 | 0.299 | 0.827 | 3 |
| 7 | 1.0 | 1500 | 15 | 9 | 10.15 | 0.342 | 0.593 | 15 |
| 8 | 1.0 | 1500 | 30 | 3 | 9.78 | 0.579 | 0.481 | 25 |
| 9 | 1.0 | 1500 | 45 | 6 | 10.25 | 0.453 | 0.521 | 21 |
| 10 | 1.5 | 500 | 15 | 6 | 10.25 | 0.198 | 0.807 | 4 |
| 11 | 1.5 | 500 | 30 | 9 | 5.08 | 0.207 | 0.907 | 1 |
| 12 | 1.5 | 500 | 45 | 3 | 9.78 | 0.292 | 0.649 | 9 |
| 13 | 1.5 | 1000 | 15 | 9 | 25.38 | 0.404 | 0.407 | 27 |
| 14 | 1.5 | 1000 | 30 | 3 | 7.34 | 0.287 | 0.706 | 5 |
| 15 | 1.5 | 1000 | 45 | 6 | 10.25 | 0.417 | 0.539 | 19 |
| 16 | 1.5 | 1500 | 15 | 3 | 11.41 | 0.386 | 0.540 | 18 |
| 17 | 1.5 | 1500 | 30 | 6 | 9.39 | 0.326 | 0.621 | 11 |
| 18 | 1.5 | 1500 | 45 | 9 | 9.02 | 0.414 | 0.564 | 16 |
| 19 | 2.0 | 500 | 15 | 9 | 10.15 | 0.328 | 0.606 | 12 |
| 20 | 2.0 | 500 | 30 | 3 | 12.23 | 0.406 | 0.515 | 22 |
| 21 | 2.0 | 500 | 45 | 6 | 5.12 | 0.4 | 0.671 | 6 |
| 22 | 2.0 | 1000 | 15 | 3 | 12.23 | 0.485 | 0.476 | 26 |
| 23 | 2.0 | 1000 | 30 | 6 | 7.68 | 0.312 | 0.670 | 7 |
| 24 | 2.0 | 1000 | 45 | 9 | 5.92 | 0.399 | 0.646 | 10 |
| 25 | 2.0 | 1500 | 15 | 6 | 13.66 | 0.283 | 0.604 | 13 |
| 26 | 2.0 | 1500 | 30 | 9 | 11.00 | 0.512 | 0.483 | 24 |
| 27 | 2.0 | 1500 | 45 | 3 | 9.78 | 0.456 | 0.527 | 20 |

## Micrographs of worn pins

Figs 11 and 12 display the SEM image of worn LM6 alloy pins and their composites with reduced wear rate and appropriate variables. Smaller grooves and broad patches are seen in the SEM pictures of the worn surface of the LM6 virgin aluminium alloy (Fig 11), suggesting a higher wear rate, most likely as a result of heat generated by friction. The interface plays a crucial role in determining the overall properties of metal-matrix composites [56–58]. Strengthening by the hard particle reinforcement depends critically on the interfacial bond. A well-bonded interface facilitates the efficient transfer and distribution of load from the matrix to the reinforcing phase, which leads to improved composite strengths [59]. Conversely, load transfer becomes less effective with a weak bond, limiting the amount of strengthening that can take place. Therefore, it is concluded that a strengthened interfacial bond is responsible for the improvement in mechanical properties. The interface also plays a key role in the fracture behaviour of metal-matrix composites. It is clear that the bond strength is related to the nature of the interface between the matrix and the reinforcement.

LM6+6% hybrid (Fig 12) composites show lesser wear rate compared to 3% & 9% hybrid composites. The less severe worn surfaces are caused by hard B$_4$C, which exhibits direct contact among the continuous phase and the more resilient

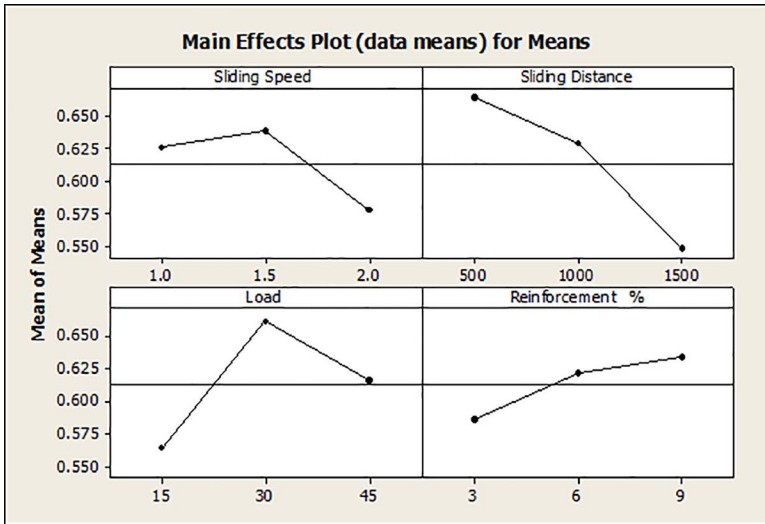

**Fig 10. Response graphs for GRG.**

**Table 5. Response table for GRG.**

| Level | Sliding Speed | Sliding Distance | Load | Reinforcement |
|---|---|---|---|---|
| 1 | 0.626 | **0.664** | 0.564 | 0.586 |
| 2 | **0.638** | 0.629 | **0.661** | 0.621 |
| 3 | 0.578 | 0.548 | 0.616 | **0.634** |
| Delta | 0.060 | 0.116 | 0.097 | 0.048 |
| Rank | 3 | 1 | 2 | 4 |

**Table 6. ANOVA for GRG.**

| Source of Variation | DOF | Sum of squares | Mean sum of squares | $F_0$ | P | Contribution (%) |
|---|---|---|---|---|---|---|
| S | 2 | 0.018 | 0.009 | 2.17 | 0.177 | 4.25% |
| D | 2 | 0.063 | 0.032 | 7.52 | 0.015 | 14.86% |
| L | 2 | 0.042 | 0.021 | 5.01 | 0.039 | 9.91% |
| SD | 4 | 0.111 | 0.028 | 6.60 | 0.012 | 26.18% |
| SL | 4 | 0.046 | 0.012 | 2.72 | 0.107 | 10.85% |
| DL | 4 | 0.110 | 0.028 | 6.52 | 0.012 | 25.94% |
| Error(Pooled) | 8 | 0.034 | 0.004 | | | 8.02% |
| Total | 26 | 0.424 | | | | 100% |

interface material. The presence of fly ash decreases the wear rate. Wear rate of composite decreased with an increase in reinforcement particles. Decrease in wear rate of composites was due to the ceramic particles acting as a load-bearing constituent, restricting deformation of the aluminium alloy. It can be concluded that the particle size is an important factor influencing wear rate. Increase in size of the ceramic particles decreased the wear rate of composites. This may be attributed to the fact that the probability of ceramic particles pulling out from aluminium will be higher in composite with small particles. However, in the case of composites with coarse particles, reinforcement remain embedded in aluminium

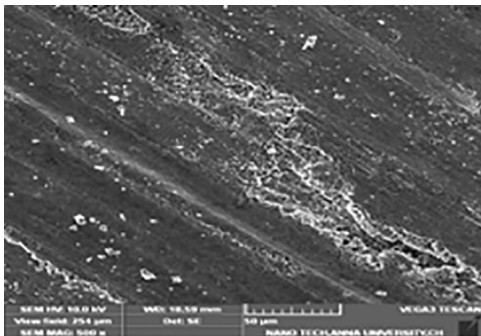

**Fig 11. Micrograph of worn pin of LM6 alloy.**

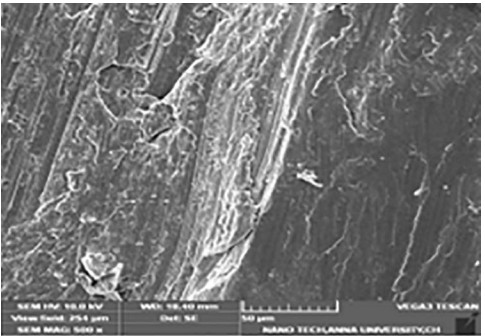

**Fig 12. Micrograph of worn pin of LM6** +6% Hybrid.

until breaking down into smaller particles may be the reason for decreased wear. Increase in hardness was also the reason for decrease in wear rate. Sudarshan and Surappa reported that aluminium composites with narrow ceramic particles exhibit superior wear resistance than composites with broad ceramic particles. In summary, composites reinforced with larger particle size offer higher wear resistance. Wear resistance also increased with increase in reinforcement percentage for all particle size ranges [60].

## Comparison of wear results

The wear results are discussed in comparison with other researches based on the input variables 'S','D', 'L' and 'R'.

**Effect of sliding speed on GRG.** Fig 13 shows the effect of sliding speed on GRG. For the tested range, raising the 'S' results in a reduction in sliding wear. Higher interfacial temperatures have been linked to a greater degree of oxidation of the Al alloy. By reducing the wear rate, the larger amount of oxide that remains serve to maintain the sliding interfaces. This could also be the cause of the current case's drop-in wear rate when 'S' increases. [61]. As 'S' rises, a shorter period will be available for debris to get away from the deteriorated track. This will suggest, subsequently, that the tribo-layer thickness grows and becomes substantially thicker over longer durations as the 'S' increases. when it turns out, when 'S' increases, the rate of wear and the CoF decreases. It has been observed that a higher sliding velocity lowers the CoF and wear rate for both MMCs. It is expected that the dispersion layer on MMC diminishes the wear rate and CoF while serving as a protective coating. The wear rate diminishes as the 'S' rises [62].

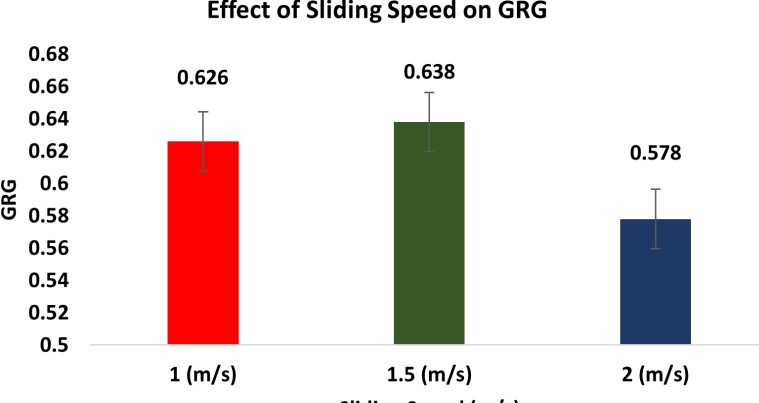

**Fig 13. Effect of Sliding Speed on GRG.**

It is widely accepted that the rate of indentation has an impact on a material's hardness. The strain rate will rise with an increase in 'S', increasing the hardness or flow strength as a result. The true area of contact will be less as a result of this raise in flow strength, which will minimise the wear rate. An additional significant consequence of increasing 'S' is the temperature rise brought on by frictional heat, which softens the material. Consequently, a greater wear rate will arise from the enhanced real area of contact [63].

**Effect of sliding distance on GRG.** Fig 14 shows the effect of sliding distance on GRG. The SWR increases as the 'D' rises. The wear volume loss rises as the 'D' does as well. This might be because increasing the 'D' does not reduce the cutting capacity of the broken particles confined among the pin and the counter face. In unreinforced alloys, adhesive wear predominates, whereas in composites, abrasive wear does. Subsurface separation is the primary mechanism in both alloys and composites at increasing loads [64].

In general, an unaltered tribo-layer is linked to a composite material with higher bonding wear resistance. The amount of 'L', 'S', operating temperature, and MMC composition all affect how this layer forms. It seems that the volume proportion of reinforcement is an important factor. Achieving a reduced counter face wear rate would need the use of materials with high hardness. It is obvious that the predicted wear rates for the counter face and composite materials must be balanced.

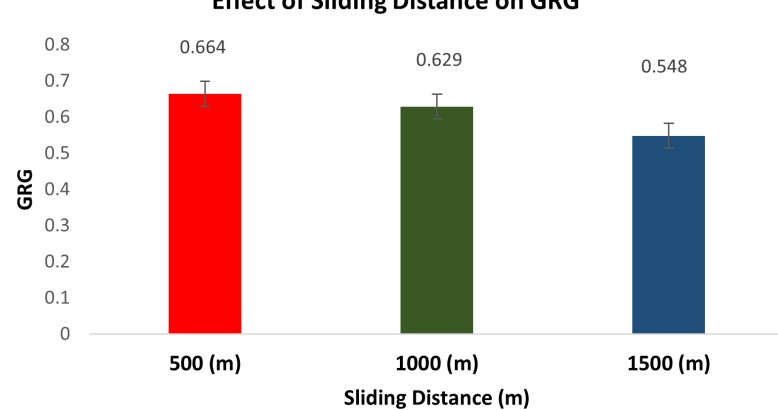

**Fig 14. Effect of Sliding Distance on GRG.**

**Effect of load.** Fig 15 shows the effect of load on GRG. It is found that when the 'L'raises, the SWR for both the composite and the LM6 alloy falls. The outcomes are consistent with those of Prakash et al [63]. The wear rate improves as the load does because the shattered particles' capacity to penetrate increases with the increase in load. The material on the surface of the pin is removed by a three-body abrasion created by the broken tiny ceramic particles that are sandwiched among the pin and the counter face [64].

Multiple investigators have documented on how 'L' influences MMC wear rate. The rate of wear of composite rises as the stress that is applied goes up, as demonstrated by all of these investigations [65].

**Effect of reinforcement percentage on GRG.** Fig 16 shows the effect of reinforcement percentage on GRG. At low sliding velocities, the rate of wear of the Hybrid composites and LM6 alloy were comparable, and the reinforcing failed to have any effect on either one. The type of dispersoid selected had no effect on the specimen's wear rates in steady state sliding. As the % of dispersoid rises, the wear rate diminishes. The pin's Al will bond with the counter face's ploughed surface, forming $Fe_2O_3$, and the ceramics will be broken down into small fragments to produce a layer known as a mechanically mixed layer (MML). The MML forms a layer among the work hardened pin and the counter face. Since the extra dispersoid will only increase the thickness of MML, durability against wear rises in smaller quantities as the dispersoid increases from 3 to 9 wt. %.

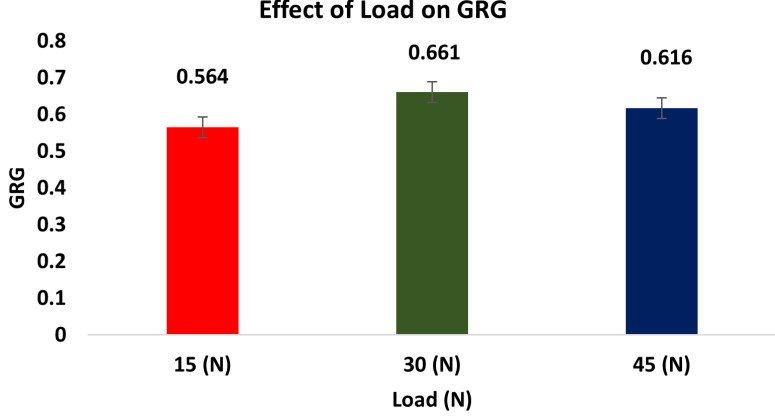

**Fig 15. Effect of Load on GRG.**

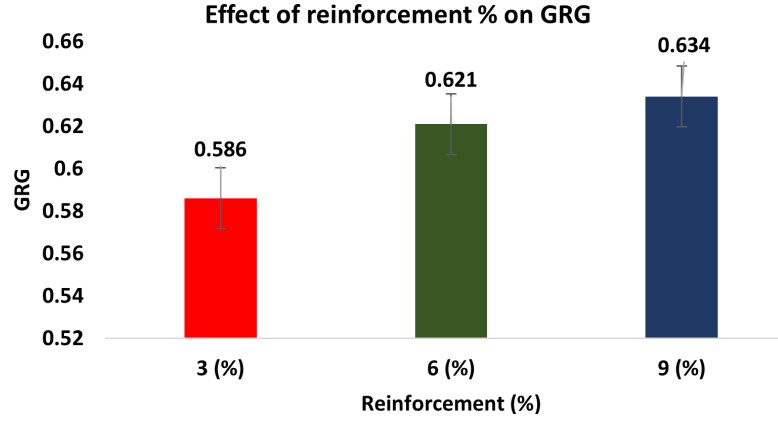

**Fig 16. Effect of Reinforcement Percentage on GRG.**

The kind, size, and rate of reinforcing particles have a significant impact on the wear of MMC. The composites had high coefficients of friction if the rate of dispersoid is low. The composite wear resistance rises steadily as the volume % of ceramic particles increases if the dispersoid is firmly attached to the continuous phase [61–63]. As the fly ash content has increased, so too has the MMCs' resistance to wear. This is because the material's overall bulk hardness is impacted by the hard fly ash particles that are present. The wear on the MMCs with low fly ash weight fractions was significant and increased approximately proportionally with period [64].

**Influence of parameters on CoF.** COF raises with raise in 'L' for LM6 matrix composites and alloy. Although there is less transfer layer forming at the point of contact, the CoF is seen at low sliding velocities and applied loads. A more rapid moving film forms at larger sliding velocities. Though, the transfer film forms and dissolves more quickly at greater loads and velocities, which lowers the friction coefficient. The raise in 'L' and speed resulted in a decrease in the value of the CoF for a given % of fly ash [65], wherein the CoF dropped as the load rose. In the final analysis, as fly ash %, weight, and speed increased, the CoF reduced as well. Both the LM6 alloy and the composite exhibits a decreasing CoF as the 'L' raises. The reason for the drop in CoF as the 'L' rises might be because more wear debris fragments are emerging from the wear layer and occupying in the spaces that exist between ceramics, which lowers the effective depth of penetration. The effective depth of penetration is decreased as a consequence of worn debris building up in the voids left by the scratches. Because of the strong ceramic particles that offer durability against abrasion and improve dry sliding wear efficiency, the CoF reduces as the 'D' rises.

## Conclusions

Three different wt. % of hybrid composites with uniform dispersion of reinforcement were created with the help of stir casting method.

- The LM6/9% hybrid is light in weight and possesses high hardness, according to measurements of density and hardness.

- The COF confirmation test reveals the smallest discrepancy (0.264) between the particular experimented value and what is expected (0.248).

- The responses have a narrow margin of error, according to confirmation studies. There exists a good agreement between them.

- From the Taguchi approach, it is found that the SD (26.18%) has the highest impact on wear rate and CoF. followed by DL, D, SL, L and S. It is seen from the ANOVA table, the contribution of SD, DL, D, SL, L and S are 26.18%, 25.94%, 14.86%, 10.85%, 9.91% and 4.25% respectively in the analysis.

The hybrid composite that was created is novel, and in the future, numerous additional combinations and optimisation strategies could potentially be employed. Owing of the ceramic reinforcements, the mechanism of wear first shifts from two-body adhesive to abrasive wear. The dispersoid lowers the wear rate and raises the composites' hardness.

## Author contributions

**Conceptualization:** Sunder Jebarose Juliyana.

**Data curation:** Charles Sarala Rubi, Sunder Jebarose Juliyana.

**Formal analysis:** Charles Sarala Rubi, Sunder Jebarose Juliyana, Emad Abouel Nasr.

**Funding acquisition:** Emad Abouel Nasr.

**Investigation:** Sachin Salunkhe, Charles Sarala Rubi, Sunder Jebarose Juliyana, Robert Cep.

**Methodology:** Sachin Salunkhe, Charles Sarala Rubi, Jayavelu Udaya Prakash, Sunder Jebarose Juliyana, Robert Cep.

**Project administration:** Sachin Salunkhe, Charles Sarala Rubi, Sunder Jebarose Juliyana, Robert Cep.

**Resources:** Sachin Salunkhe, Charles Sarala Rubi, Sunder Jebarose Juliyana, Emad Abouel Nasr.

**Software:** Sachin Salunkhe, Jayavelu Udaya Prakash, Emad Abouel Nasr.

**Supervision:** Jayavelu Udaya Prakash, Emad Abouel Nasr.

**Validation:** Sachin Salunkhe, Jayavelu Udaya Prakash, Robert Cep, Emad Abouel Nasr.

**Visualization:** Jayavelu Udaya Prakash, Robert Cep.

**Writing – original draft:** Jayavelu Udaya Prakash, Sunder Jebarose Juliyana, Robert Cep, Emad Abouel Nasr.

**Writing – review & editing:** Jayavelu Udaya Prakash, Sunder Jebarose Juliyana, Emad Abouel Nasr.

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
