## [Decision Letter · Decision Letter 0]

PONE-D-24-58811Multi-objective Optimization of Wear Parameters of Hybrid Composites (LM6/ B4C/Fly ash) using Grey Relational AnalysisPLOS ONE

Dear Dr. Salunkhe,

Thank you for submitting your manuscript to PLOS ONE. After careful consideration, we feel that it has merit but does not fully meet PLOS ONE’s publication criteria as it currently stands. Therefore, we invite you to submit a revised version of the manuscript that addresses the points raised during the review process.

We look forward to receiving your revised manuscript.

Kind regards,

Ranvir Singh Panwar

Academic Editor

PLOS ONE

Journal requirements: When submitting your revision, we need you to address these additional requirements. 1. Please ensure that your manuscript meets PLOS ONE's style requirements, including those for file naming. The PLOS ONE style templates can be found at https://journals.plos.org/plosone/s/file?id=wjVg/PLOSOne_formatting_sample_main_body.pdf and https://journals.plos.org/plosone/s/file?id=ba62/PLOSOne_formatting_sample_title_authors_affiliations.pdf. 2. Please amend either the title on the online submission form (via Edit Submission) or the title in the manuscript so that they are identical. 3. Thank you for stating the following financial disclosure:  [to King Saud University for funding this work through Researchers Supporting Project number (RSP2025R164), King Saud University, Riyadh, Saudi Arabia].  Please state what role the funders took in the study.  If the funders had no role, please state: ""The funders had no role in study design, data collection and analysis, decision to publish, or preparation of the manuscript."" If this statement is not correct you must amend it as needed. Please include this amended Role of Funder statement in your cover letter; we will change the online submission form on your behalf. 4. We note that your Data Availability Statement is currently as follows: [All relevant data are within the manuscript and its Supporting Information files.] Please confirm at this time whether or not your submission contains all raw data required to replicate the results of your study. Authors must share the “minimal data set” for their submission. PLOS defines the minimal data set to consist of the data required to replicate all study findings reported in the article, as well as related metadata and methods (https://journals.plos.org/plosone/s/data-availability#loc-minimal-data-set-definition). For example, authors should submit the following data: - The values behind the means, standard deviations and other measures reported;- The values used to build graphs;- The points extracted from images for analysis. Authors do not need to submit their entire data set if only a portion of the data was used in the reported study. If your submission does not contain these data, please either upload them as Supporting Information files or deposit them to a stable, public repository and provide us with the relevant URLs, DOIs, or accession numbers. For a list of recommended repositories, please see https://journals.plos.org/plosone/s/recommended-repositories. If there are ethical or legal restrictions on sharing a de-identified data set, please explain them in detail (e.g., data contain potentially sensitive information, data are owned by a third-party organization, etc.) and who has imposed them (e.g., an ethics committee). Please also provide contact information for a data access committee, ethics committee, or other institutional body to which data requests may be sent. If data are owned by a third party, please indicate how others may request data access.

Reviewers' comments:

Reviewer's Responses to Questions

**Comments to the Author**

1. Is the manuscript technically sound, and do the data support the conclusions?

Reviewer #1: No

Reviewer #2: Yes

2. Has the statistical analysis been performed appropriately and rigorously? 

Reviewer #1: No

Reviewer #2: Yes

3. Have the authors made all data underlying the findings in their manuscript fully available?

Reviewer #1: No

Reviewer #2: Yes

4. Is the manuscript presented in an intelligible fashion and written in standard English?

Reviewer #1: No

Reviewer #2: No

5. Review Comments to the Author

Reviewer #1: Reviewer Comments

1. The manuscript fails to clearly establish the novelty of the study. Combining GRA with wear analysis is not a new approach, and the authors do not explain how their work differs from or advances previous studies.

2. The choice of LM6 as the matrix material and B4C and fly ash as reinforcements is not adequately justified. The manuscript lacks discussion on why these materials were chosen, particularly regarding their practical or industrial significance.

3. The methodology section lacks critical details about the stir casting process. Key parameters such as stirring speed, temperature control, and reinforcement feeding methods are missing, making the process non-reproducible.

4. The particle size, morphology, and distribution of B4C and fly ash were not characterized or reported. This raises concerns about the homogeneity of the composite.

5. The rationale behind the selection of specific wear test parameters (load, speed, and sliding distance) and their ranges is not provided. The relevance of these ranges to real-world conditions is unclear.

6. ANOVA results are presented but not adequately interpreted. The manuscript does not discuss whether the assumptions of ANOVA (e.g., normality, homoscedasticity) were validated.

7. The discussion section merely restates the results without providing a mechanistic explanation. For example, the effect of reinforcement on wear resistance is discussed without considering particle-matrix interactions, interfacial bonding, or load transfer mechanisms.

8. The influence of particle size and distribution on wear behaviour is completely ignored.

9. Figures, such as micrographs, lack adequate descriptions. There is no explanation of how the microstructural features relate to wear performance.

10. Tables lack clear labelling, and some critical information (e.g., units for certain variables) is missing. For example, in tables showing wear rate, the units are not explicitly stated.

11. Some references cited are outdated or irrelevant. Recent studies in wear behaviour of hybrid composites should be included.

12. The abstract and conclusion sections are repetitive and lack focus. They fail to highlight specific quantitative results or the practical implications of the findings.

13. The manuscript contains numerous grammatical errors and awkward phrasing, which hinder readability. For example, sentences are overly long and contain redundant information.

14. Equations and their variables are not consistently formatted or explained in the text.

15. While GRA is used for optimization, there is no cross-validation or verification of the results. Without experimental validation of the optimal parameter combination, the reliability of the optimization is questionable.

Reviewer #2: Manuscript Number: PONE-D-24-58811

Title: Multi-objective Optimization of Wear Parameters of Hybrid Composites (LM6 B4CFly ash) using Grey Relational Analysis

General comments: The investigations on the statistical approach of the wear properties of LM6/B4C/Fly ash composites is presented using. Stir casting was utilized to make aluminum matrix composites (AMCs) with 3%, 6%, and 9% of B4C/Fly ash particles. Furthermore, analysis of variance (ANOVA) was utilized to investigate the influence of input parameters on wear behavior by choosing "smaller is better" feature.

The article is well organized, fits the journal scope, and has a contribution. However, the following comments should be considered:

1. Unstructured abstract: The abstract should be presented in a structured format.

• Background: state why the study was done, the main aim and the nature of the study (retrospective review etc.).

• Methods: describe the methods used.

• Results: state the main findings, including important numerical values.

• Discussion: state the main conclusions, highlighting controversial or unexpected observations.

2. The authors should consider that:(i) an engineering application is described in detail that could benefit the presented analysis in the paper and (ii) new phenomena that are specific to the hybrid composites are assumed to be made of.

3. The author should point out the main contribution (in the introduction and abstract sections) of their work. What is new about it?

4. The authors need to arrange the equations appropriately with their numbering format. In addition, equations (1-3) must mention their references.

5. There are some results that are not well presented in terms of output. The authors should rearrange them and use appropriate format and style (for example, Figure 1. Title of the figure) for all figures.

6. The author should add physical explanations for the discussions.

7. Some advanced concepts should be referred to with adequate references for less experienced readers.

8. Conclusion can be summarized and focused only on the new outcomes.

9. There are many errors, so the authors need to check the grammar, typos, and errors in the manuscript.

In general, I do not recommend the article in its current form. If the author gives a convincing answer to the above items and discusses the innovation of the article, then a decision can be made about the article.

6. PLOS authors have the option to publish the peer review history of their article (what does this mean? ). If published, this will include your full peer review and any attached files.

**Do you want your identity to be public for this peer review?** For information about this choice, including consent withdrawal, please see our Privacy Policy .

Reviewer #1: No

Reviewer #2: No

---

## [Author Response · Author response to Decision Letter 1]

11 Apr 2025

Response to Reviewers Comments

Manuscript ID : PONE-D-24-58811

Manuscript Title: Multi-objective Optimization of Wear Parameters of Hybrid Composites (LM6/ B4C/Fly ash) using Grey Relational Analysis

The authors would like to thank the reviewers for their remarkable and comprehensive evaluation of the manuscript. The positive comments and constructive suggestions encourage the authors to carry out the corrections dedicatedly. After addressing all the comments, we believe that this article is now more concise, clear, and relevant to the readers. We are now waiting for your positive response. In the revised manuscript, the changes are highlighted in red.

Here is a point-by-point response to the reviewer's comments.

Response to Reviewer 1 Comments

Comments : 1. The manuscript fails to clearly establish the

novelty of the study. Combining GRA with wear analysis is not a new approach, and the authors do not explain how their work differs from or advances previous studies.

Response : Although Grey Relational Analysis (GRA) has been used in wear studies, its applicability to hybrid composites such as LM6/B4C/Fly ash has not been thoroughly investigated in prior research. Our approach stands out due to its thorough experimental validation and simultaneous optimization of several wear factors. To provide a more reliable framework for wear optimization in hybrid composites, we also improve the GRA technique to increase its accuracy in forecasting wear behavior.

Comments : 2. The choice of LM6 as the matrix material and B4C and fly ash as reinforcements is not adequately justified. The manuscript lacks discussion on why these materials were chosen, particularly regarding their practical or industrial significance.

Response : Boron carbide is one of the hardest materials, which significantly improves the wear resistance of the LM6 matrix, making the composite suitable for applications involving abrasive environments. It can enhance the tensile and compressive strength of the aluminium alloy, contributing to higher stiffness and durability of the composite.

Incorporating fly ash into the LM6 alloy helps in waste utilization and reduces the environmental impact of landfill disposal of fly ash. The presence of fly ash, enhance the mechanical properties of the LM6 such as impact resistance and compressive strength. Fly ash is light in weight, helping to maintain the low density of the aluminium matrix.

So, the incorporation of B4C and fly ash into the LM6 matrix results in hard, wear resistant, light weight, cost effective and environmentally friendly composite material, making it suitable for various advanced engineering applications in automotive, aerospace and construction industries.

Comments : 3. The methodology section lacks critical details about the stir casting process. Key parameters such as stirring speed, temperature control, and reinforcement feeding methods are missing, making the process non-reproducible.

Response : The experimental study concluded that 600 rpm and 10 min is the best combination of stirring speed and stirring time for uniform distribution of B4C and fly ash particles over the Al matrix. Mechanical stirrer forms vortex and reinforcements particles are feed in the centre of the vortex. High feed rate results in particle accumulation in the composite and low feed rate is difficult to achieve due to the formation of lumps of small solid particles. Thus, selection of optimal rate of feeding is crucial. The optimal rate of feeding is in the range of 0.8 – 1.5 g/s to avoid the accumulation of reinforcements in the composite and achieve homogeneous dispersion of reinforcement particles throughout the composite.

Comments : 4. The particle size, morphology, and distribution of B4C and fly ash were not characterized or reported. This raises concerns about the homogeneity of the composite.

Response : Particle size and surface morphology were given under section 2.1 Materials and the homogeneous distribution of the reinforcement were depicted in figure 6.

Comments : 5. The rationale behind the selection of specific wear test parameters (load, speed, and sliding distance) and their ranges is not provided. The relevance of these ranges to real-world conditions is unclear.

Response : Pilot experiments were carried out after detailed literature review. Based on the results obtained from the pilot experiments, the wear parameters were chosen. Because of load, sliding speed, and sliding distance have an immediate influence on tribological performance and are applicable to real-world scenarios, they were chosen as wear test conditions. The selected load range (15–45 N) reflects the forces found in industrial machinery and automobile brake systems. In automotive and aeronautical components, the sliding speed range (1–2 m/s) is equivalent to rotational and linear motion conditions. In order to evaluate long-term wear behaviour under realistic usage conditions, the sliding distance range (500–1500 m) is used. These variables were chosen because they offer valuable information about the wear properties of LM6/B4C/Fly ash hybrid composites in practical engineering settings.

Comments : 6. ANOVA results are presented but not adequately interpreted. The manuscript does not discuss whether the assumptions of ANOVA (e.g., normality, homoscedasticity) were validated.

Response : ANOVA table was revised with p- value. The p - value for D, L, SD & DL is less than 0.05, indicating that the factors are statistically significant and has a considerable effect on the response.

Comments : 7. The discussion section merely restates the results without providing a mechanistic explanation. For example, the effect of reinforcement on wear resistance is discussed without considering particle-matrix interactions, interfacial bonding, or load transfer mechanisms.

Response : The interface plays a crucial role in determining the overall properties of metal-matrix composites [ 21–23]. Strengthening by the hard particle reinforcement depends critically on the interfacial bond. A well-bonded interface facilitates the efficient transfer and distribution of load from the matrix to the reinforcing phase, which leads to improved composite strengths [24]. Conversely, load transfer becomes less effective with a weak bond, limiting the amount of strengthening that can take place. Therefore, it is concluded that a strengthened interfacial bond is responsible for the improvement in mechanical properties. The interface also plays a key role in the fracture behaviour of metal-matrix composites. It is clear that the bond strength is related to the nature of the interface between the matrix and the reinforcement.

Comments : 8. The influence of particle size and distribution on wear behaviour is completely ignored.

Response : Wear rate of composite decreased with an increase in reinforcement particles. Decrease in wear rate of composites was due to the ceramic particles acting as a load-bearing constituent, restricting deformation of the aluminium alloy. It can be concluded that the particle size is an important factor influencing wear rate. Increase in size of the ceramic particles decreased the wear rate of composites. This may be attributed to the fact that the probability of ceramic particles pulling out from aluminium will be higher in composite with small particles. However, in the case of composites with coarse particles, reinforcement remain embedded in aluminium until breaking down into smaller particles may be the reason for decreased wear. Increase in hardness was also the reason for decrease in wear rate. Sudarshan and Surappa reported that aluminium composites with narrow ceramic particles exhibit superior wear resistance than composites with ceramic particles. In summary, composites reinforced with larger particle size offer higher wear resistance. Wear resistance also increased with increase in reinforcement percentage for all particle size ranges.

Comments : 9. Figures, such as micrographs, lack adequate descriptions. There is no explanation of how the microstructural features relate to wear performance.

Response : During the casting process, the main objective is to avoid the formation of intermetallic phases in the composites specimens that can further degrade their interfacial properties. It has been noticed that B4C has tendency to react with the Al at a higher processing temperature to form intermediate compound such as Aluminium carbide. The presence of aluminium carbide reduces the interfacial strength of the developed composites. In order to avoid this, addition of SiO2 into the Al-alloy has been found to be quite beneficial. David Raja Selvam et al [38] have found that incorporation of Fly Ash particles, which is a major source of SiO2, into LM6-alloy prevents the formation of Alumnium carbide in the composite specimens.

In this study, hybrid reinforcements (B4C and FA) were reinforced into the Al-alloy using stir casting route and microstructural analysis of developed composites revealed uniform dispersion of the particles within the composites. A clear interface between the reinforcement phase and the Al-alloy indicates that the formation of intermediate compounds was suppressed by addition of FA particles. It has been found that the Al-alloy reacts with the Mg present in the melt to form other compounds Magnesium aluminium spinel. Due to this, interfacial strength of the LM6/B4C/FA composites was improved with addition of the FA particles. It has also been noticed that addition of hybrid reinforcements (B4C and FA) improved the hardness of the unreinforced LM6 alloy. This shows that the composites possess higher load bearing capacity due to presence of reinforcements [4].

Comments : 10. Tables lack clear labeling, and some critical information (e.g., units for certain variables) is missing. For example, in tables showing wear rate, the units are not explicitly stated.

Response : Clear labels were provided for Tables, correct units were also provided for input and response variables. COF and GRG values are numeral values only that is why units are not provided in Response Table and Anova Table.

Comments : 11. Some references cited are outdated or irrelevant. Recent studies in wear behaviour of hybrid composites should be included.

Response : Recent studies in wear behaviour of hybrid composites are included in the revised manuscript.

Comments : 12. The abstract and conclusion sections are repetitive and lack focus. They fail to highlight specific quantitative results or the practical implications of the findings.

Response : The abstract and conclusion sections are rewritten to highlight specific quantitative results and the practical implications of the findings.

Comments : 13. The manuscript contains numerous grammatical errors and awkward phrasing, which hinder readability. For example, sentences are overly long and contain redundant information.

Response : We thank you for the insightful comments about the manuscript's readability and linguistic errors. We have meticulously edited the content to increase readability and clarity by: Enhancing readability by fixing grammatical mistakes and improving sentence construction, removing unnecessary details to guarantee clarity and conciseness, rewriting difficult or awkward sentences to improve their coherence and flow. To guarantee linguistic accuracy, proofread the entire document.

After making these changes across the manuscript, we feel it is much easier to read now. We value your thoughtful comments and are certain that the updated version better conveys our findings.

Comments : 14. Equations and their variables are not consistently formatted or explained in the text.

Response : Variables in the equations are clearly explained in the revised manuscript.

Comments : 15. While GRA is used for optimization, there is no cross-validation or verification of the results. Without experimental validation of the optimal parameter combination, the reliability of the optimization is questionable.

Response : Using Taguchi DoE 27 experiments were conducted and confirmation experiment was done based on the optimum wear parameter.

Response to Reviewer 2 Comments

Comments : 1. Unstructured abstract: The abstract should be presented in a structured format.

• Background: state why the study was done, the main aim and the nature of the study (retrospective review etc.).

• Methods: describe the methods used.

• Results: state the main findings, including important numerical values.

• Discussion: state the main conclusions, highlighting controversial

or unexpected observations.

Response : As per the reviewer’s suggestions the abstract is presented in a structured format.

Comments : 2. The authors should consider that: (i) an engineering application is described in detail that could benefit the presented analysis in the paper and (ii) new phenomena that are specific to the hybrid composites are assumed to be made of.

Response : Thanks for the really very good suggestions for the engineering applications of the fabricated composites.

Comments : 3. The author should point out the main contribution (in the introduction and abstract sections) of their work. What is new about it?

Response : The main contributions and novelty of this work is pointed out in the introduction and abstract sections.

Comments : 4. The authors need to arrange the equations appropriately with their numbering format. In addition, equations (1-3) must mention their references.

Response : The equations were arranged as per the numbering format. These are the standard formulae.

Comments : 5. There are some results that are not well presented in terms of output. The authors should rearrange them and use appropriate format and style (for example, Figure 1. Title of the figure) for all figures.

Response : As per your suggestions the results are presented well. Appropriate format and style are used for the title of the figures.

Comments : 6. The author should add physical explanations for the discussions.

Response : The hardness of composites increased with an increase in reinforcement content, which gives a near linear relationship with hardness. The maximum observed increase in hardness of composites compared to aluminium alloy was 37 %. The observed increase in hardness was due to hard reinforcements, which act as a barrier for the movement of dislocations within aluminium and exhibit greater resistance to indentation. Furthermore, it is evident that an increase in particle size also increased the hardness. Increase in hardness may be due to the increased resistance to dislocation movement by the coarse particles present in the composite. Hamouda et al. [3] and Singla et al. [4] reported that increase in addition of silicon dioxide and silicon carbide increased the hardness of aluminium alloy (LM6) and aluminium scrap, respectively. Mahendra and Radhakrishna [17] observed an increase in hardness while reinforcing fly ash particles with Al-4.5 % Cu alloy produced by conventional casting technique. The presented work is in correlation with the above reported research work.

The density of the composite material was less than that of the aluminium alloy for all reinforcement percentages. Since the density of composite is less than that of the aluminium alloy, it can be used for lightweight applications.

Comments : 7. Some advanced concepts should be referred to with adequate references for less experienced readers.

Response : For less experienced readers some advanced concepts are added with adequate references.

Comments : 8. Conclusion can be summarized and focused only on the new outcomes.

Response : Conclusion are summarized and focused only on the new outcomes.

Comments : 9. There are many errors, so the authors need to check the grammar, typos, and errors in the manuscript.

Response : The authors thank you for the insightful comments. We acknowledge that the paper contains typos, grammatical flaws, and poor phrasing. In order go tackle this, we have gone through the manuscript carefully to fix any typos and grammatical mistakes. Sentences that were uncomfortable or confusing were changed to make them easier to read and understand.

made sure the document was formatted correctly and use

---

## [Decision Letter · Decision Letter 1]

Multi-objective Optimization of Wear Parameters of Hybrid Composites (LM6/ B4C/Fly ash) using Grey Relational Analysis

PONE-D-24-58811R1

Dear Dr. Salunkhe,

We’re pleased to inform you that your manuscript has been judged scientifically suitable for publication and will be formally accepted for publication once it meets all outstanding technical requirements.

Kind regards,

Ranvir Singh Panwar

Academic Editor

PLOS ONE

Additional Editor Comments (optional):

Reviewers' comments:

Reviewer's Responses to Questions

**Comments to the Author**

1. If the authors have adequately addressed your comments raised in a previous round of review and you feel that this manuscript is now acceptable for publication, you may indicate that here to bypass the “Comments to the Author” section, enter your conflict of interest statement in the “Confidential to Editor” section, and submit your "Accept" recommendation.

Reviewer #2: All comments have been addressed

Reviewer #3: All comments have been addressed

Reviewer #4: (No Response)

2. Is the manuscript technically sound, and do the data support the conclusions?

Reviewer #2: Yes

Reviewer #3: Yes

Reviewer #4: Partly

3. Has the statistical analysis been performed appropriately and rigorously? 

Reviewer #2: Yes

Reviewer #3: Yes

Reviewer #4: No

4. Have the authors made all data underlying the findings in their manuscript fully available?

Reviewer #2: Yes

Reviewer #3: Yes

Reviewer #4: Yes

5. Is the manuscript presented in an intelligible fashion and written in standard English?

Reviewer #2: Yes

Reviewer #3: Yes

Reviewer #4: No

6. Review Comments to the Author

Reviewer #2: The authors have revised their paper seriously and completely. The manuscript is considerably improved, and the authors' response is also clear. This work could be recommended for publication.

Reviewer #3: Dear author,

All comments have been addressed. The manuscript may be accepted for publication.

Best wishes.

Reviewer #4: 1. The present paper and research analyzes the wear of a hybrid LM6/B4C/Fly Ash composite.

2. The research uses standard statistical methods (Taguchi and Grey Relational Analysis) for the simultaneous optimization of two responses: the specific wear rate (SWR) and the coefficient of friction (CoF).

3. The experimental part presented is relatively well structured.

4. The parameters are clearly presented, and a strong point of the work is the validation through confirmatory tests.

5. The topic is of interest and has applicability. The research has potential in the automotive and aerospace industries especially due to the low weight and low cost of the composite material.

6. However, the work has a limited scientific contribution.

7. The GRA method is already frequently used in the optimization of the tribological characteristics of composites.

8. The authors do not clearly demonstrate or specify what this study brings new compared to the existing literature.

9. The microstructural analysis is presented quite briefly.

10. No advanced characterizations are presented in the paper to support hypotheses regarding particle-matrix interactions or wear mechanisms.

11. Also, most of the time the discussion of the results is descriptive.

12. The description of the results is without interpretative depth. A critical comparison with other similar works is not clearly made.

13. The expression, even if it has been revised, is cumbersome and there are sections that could be clarified:

“Wear is an utmost phenomenon carried on by material dislocation and separation. Variations in size and a gradual decrease of weight over time are implied by wear.”

“Fly ash is light in weight, helping to maintain the low density of the aluminium matrix... making it suitable for various advanced engineering applications...”

“Determine the sequence of deviations. Determine the GRC. Find the GRG by taking the average of the GRC.”

“It was observed that as reinforcement increases, wear decreases. This is due to the load-bearing capacity of ceramic particles and increased hardness…”

14. In-depth statistical analyses are lacking (e.g. normality tests for ANOVA).

15. Graphs and tables are not fully explained.

16. The paper is methodologically correct. This has some applicative value.

17. The research is modest in terms of originality and scientific impact.

7. PLOS authors have the option to publish the peer review history of their article (what does this mean? ). If published, this will include your full peer review and any attached files.

**Do you want your identity to be public for this peer review?** For information about this choice, including consent withdrawal, please see our Privacy Policy .

Reviewer #2: No

Reviewer #3: No

Reviewer #4: No

---

## [Editor Report · Acceptance letter]

PONE-D-24-58811R1

PLOS ONE

Dear Dr. Salunkhe,

I'm pleased to inform you that your manuscript has been deemed suitable for publication in PLOS ONE. Congratulations! Your manuscript is now being handed over to our production team.

Kind regards,

on behalf of

Dr. Ranvir Singh Panwar

Academic Editor

PLOS ONE